# Comprehensive Analysis of Mechanical Properties of CB/SiO_2_/PVDF Composites

**DOI:** 10.3390/polym12010146

**Published:** 2020-01-07

**Authors:** Fangyun Kong, Mengzhou Chang, Zhenqing Wang

**Affiliations:** 1College of Aerospace and Civil Engineering, Harbin Engineering University, Harbin 150001, China; kongfangyun@hrbeu.edu.cn; 2College of Equipment Engineering, Shenyang Ligong University, Shenyang 110000, China; changmengzhou@hrbeu.edu.cn

**Keywords:** CB/SiO_2_/PVDF composite materials, uniaxial tensile test, mechanical properties, microstructure characterization

## Abstract

Damage is a key problem that limits the application of polymer membranes. In this paper, conductive carbon black (CB) and silicon dioxide (SiO_2_)-reinforced polyvinylidene fluoride (PVDF) composites were prepared using a solution mixing method. Through a uniaxial tensile test, the fracture and damage characteristics of the material were analyzed. When the structure had inevitable notch damage, changing the notch angle was very helpful for the material to bear more load. In addition, when there were two kinds of fillers in the PVDF matrix at the same time, there was an interaction between particles. The microstructure of the composite was characterized by scanning electron microscopy (SEM), energy-dispersive spectrometry (EDS), and thermogravimetric (TG) analysis. The experimental results indicate that, when the ratio of CB:SiO_2_:PVDF was 1:4:95, the general mechanical properties of the composite were the best.

## 1. Introduction

Compared with ordinary materials, composite materials have many characteristics which can improve or overcome the weakness of a single material, giving full play to the advantages of each material. The optimum performance design of materials can be carried out according to the functional requirements. Commonly used composite matrices are metal matrices, inorganic material matrices, and polymer matrices. In contrast, polymeric materials own good formability and are lightweight, which holds great promise for applications. Polyvinylidene fluoride (PVDF) is a typical polymer material widely used in various fields of engineering applications due to its excellent physical properties. Firstly, it has good thermal and mechanical properties. At a certain pressure and temperature, it can still maintain good mechanical strength. It is mainly used for valves, pipes, and heat exchangers. Secondly, PVDF has good chemical stability and can resist the corrosion of most organic solvents, inorganic acids, and aromatic hydrocarbons. It can be used for the storage and transportation of highly purified chemicals. Thirdly, PVDF has strong weather resistance, as well as high anti-fading and anti-ultraviolet performance in harsh environments. The composite material is mainly used in building walls, roads, and airports. Finally, PVDF has five different crystal types, among which the β crystal type of PVDF has excellent dielectric and piezoelectric properties. It is mainly used for ultrasonic measurement, pressure sensors, and detonation.

In the study of modified composites, carbon black (CB) is one of the main fillers used to improve the piezoelectric and mechanical properties of composites. Batteries, sensors, and piezoelectric films all need composite materials with more stable storage, sensing, and other functions [1,2,3]. Xu et al. [4] studied the dynamic piezoresistance of a flexible nanocomposite strain sensor made of CB nanoparticles and PVDF. In the time domain analysis, the averaged strain interval (ASI) value was less than 0:8 mε, which indicates that the CB/PVDF sensor can detect extremely weak strain related to structural damage, as well as high-frequency and ultrasonic vibration; the excellent sensing accuracy and response frequency of the CB/PVDF sensor are related to the microstructure of the nanoparticles. Wu’s [5] results showed that, when the optimum CB content was 0.5 wt.%, the optimum output voltage and power density of the composite film were 204% and 464% of the original PVDF–hexafluoropropylene (HFP) film. The addition of excessive CB would lead to smaller crystal size and lower crystallinity, thus reducing the piezoelectric properties of PVDF–HFP/CB composite films. Lazarraga et al. [6] prepared lithium battery anode composites with LiMn_2_O_4_ as the electrochemical active material, CB as the conductor, and PVDF as the binder. When the CB content was 2 wt.%, the conductivity increased with the increase in percolation process. The composite exhibited elastic behavior at a stress of 6 MPa. Fuertes [7] and Ha et al. [8] prepared electrode composites with PVDF and silicon dioxide (SiO_2_) as a matrix, and CB and LiMn_2_O_4_ as a conductor. The influence of CB content on relative capacitance, energy density, and power density was studied. Adding CB nanoparticles to the PVDF matrix can improve its dielectric properties [9,10], reduce the penetration threshold of the composite, and improve the conductivity and mechanical properties of the composite [11,12].

The addition of CB nanoparticles can not only improve the piezoelectric properties, but also change the micro properties of the composite. The internal structure of the composite is more compact, and the composite has corrosion resistance and wear resistance. Brostow et al. [13] optimized the friction and wear properties of PVDF by adding CB nanoparticles. Adding a proper amount of CB lubricants (such as 3%) to PVDF composite can reduce the friction of the material. However, less (1% or 1.5%) or more particles would increase the friction of the composite. Antunes et al. [14] studied the corrosion resistance of CB–synthetic graphite (SG)–PVDF composites. It was found that the corrosion resistance decreased when the CB content increased to 5 wt.%. The highest conductivity was achieved for the composition CB = 5 wt.%, PVDF = 15 wt.%, and SG = 80 wt.%. Thomas et al. [15] studied the effects of CB temperature, soaking time, polymer solution concentration, and casting PVDF film thickness on the film properties. A new type of film with uniform surface structure and asymmetric pore structure was obtained. Sardarabadi et al. [16] prepared CB/PVDF composites with different mass fractions. It was found that the addition of CB filler made the membrane structure more compact, with a lower permeation flux and a higher contact angle and pervaporation separation index.

In engineering applications, the mechanical properties of composite materials were widely and deeply studied by scholars. Laiarinandrasana et al. [17] analyzed the mechanical behavior of PVDF. Smooth notched specimens were prepared and tested at −100 °C to 20 °C under different strain rates. A larger brittle–ductile area ratio was found in the case of higher strain rate and lower temperature. Ke et al. [18] prepared multi-walled carbon nanotube (MWCNT)/CB/PVDF composites by melt mixing. It was found that the nucleation efficiency of MWCNTs was higher than that of CB for the crystallization of PVDF. Dynamic mechanical analysis showed that MWCNTs have a greater contribution to the storage modulus of PVDF, especially at low temperature. Haddadi et al. [19] studied the effect of different nano-SiO_2_ content on the mechanical and thermodynamic properties of PVDF. The results showed that the tensile strength of the composite was 450% higher than that of PVDF when the fiber diameter was 125–350 nm and the SiO_2_ content was 2 wt.%.

Many researchers studied the mechanical properties of PVDF and its composites, including the modulus of elasticity, viscoelasticity, ultimate stress, and strain. However, in the practical application of film materials, fracture damage is inevitable. In this paper, different kinds of single-edge notched tensile tests were carried out to investigate the damage mechanism of PVDF matrix composites prepared in the laboratory. The fracture damage characteristics of the materials were systematically studied by means of scanning electron microscopy (SEM), X-ray diffraction (XRD), and energy-dispersive spectrometry (EDS).

## 2. Materials and Methods

### 2.1. Materials

The materials used in the experiment were PVDF, SiO_2_, CB, *N*,*N*-dimethylformamide (DMF, AR), and a silane coupling agent (KH550, AR). PVDF was purchased from Shanghai O-Fluorine Chemical Technology Co. Ltd., Shanghai, China. SiO_2_ with a 200-nm average diameter was purchased from Xilong Scientific Co. Ltd., Jiangsu, China. CB (30–45 nm) was purchased from XFNANO, INC., Jiangsu, China. Powder materials such as PVDF were dried in the oven before use. The chemicals were used without further purification in this work.

### 2.2. Preparation of Composite Films

The fabrication process of the composite samples was based on the method proposed in our previous work [20]. Firstly, CB was dissolved in a DMF solution at 700 rpm in a magnetic stirrer (MYP11-2, Chijiu Co., Shanghai, China). A certain amount of PVDF and silica powder was added to the liquid mixture and stirred for 3 h at 800 rpm. Secondly, the mixed solvent was dispersed for 10–15 min using an ultrasonic vibration machine. Appropriate low-power ultrasonic oscillation is beneficial to the uniform distribution of pores in the material. Strong mixing and ultrasonic shock was applied for two cycles; then, the well-mixed solution was poured into the mold for heating and drying. The prepared PVDF composite film had a smooth surface and a uniform thickness (as shown in Figure 1), and it was appropriately cut and tested.

### 2.3. Material Characterization

Uniaxial tensile tests were performed using a tensile tester (1023 Laboratory of mechanics experiment center) at room temperature. Referring to ASTM D 882-2012, the initial clamping distance was 50 mm and the total length of the sample was 150 mm. A cross-head speed of 5 mm/min was used. The detailed sample sizes are described in Section 3.1.

The surface and cross-section morphologies of the films were characterized by a Supra 55 SEM (Carl Zeiss AG, Oberkochen, Germany). The films were fractured in liquid nitrogen after 30 min of immersion and sprayed with a gold layer before being examined at different magnifications at an acceleration voltage of 5.0–10.0 kV.

The energy-dispersive EDS device was used to output the mapping and spectrum results, allowing the analysis of the composition of elements in the composite material, such as Si, O, C, F, etc.

XRD (PANalytical, Almelo, Netherlands) was used to analyze the morphological structure and surface characteristics of the films (referring to standard GB/T 17359-2012). The effects of filler type and filling amount on the crystallization behavior of the PVDF matrix in composites is discussed later.

The thermal properties of the composites were studied by thermogravimetric analysis (TG) and differential TG (DTG) according to the ASTM D7426-171 08(2013) standard. The instrument used in the experiment was an STA6000 (PerkinElmer, Waltham, MA, USA).

## 3. Results

### 3.1. Effect of Notch Geometry on Stress–Strain Relationship of SiO_2_/PVDF

In this section, the effect of the sample notch angle on the mechanical properties of composites was studied. The modified filler ratios of SiO_2_/PVDF composites prepared were as follows: 0 wt.%, 2 wt.%, 4 wt.%, 6 wt.%, and 8 wt.% (SiO_2_). The notch angles of the tested material were 60° and 120°, and the depth of each notch was a = 2 mm. The specific sample size was 10 mm × 150 mm, and the notch location was as shown in Figure 2a.

The thickness of the film samples was measured using a spiral micrometer. Each sample was measured three times to obtain the average value. The instrument used in this test was a Z010 electronic universal testing machine (Zwick), shown in Figure 2b. The distance between two fixed ends of the specimen was 50 mm, and the loading speed was 5 mm/min. In this experiment, each sample was measured five times on average. The strain–stress curves of the films were obtained from the experimental results, as shown in Figure 3. The tensile strength of the films increased with the increase in SiO_2_ content. When the content of SiO_2_ was about 6 wt.%, the tensile strength of the samples with different angle notches was the largest; however, when the filling content exceeded 6 wt.%, the tensile strength of the composites began decreasing.

In Table 1, the ultimate stress values of the single notch tensile test with different contents and different angles are listed. The experimental conditions of composite samples were the same, whereby the notch angles were 60° and 120°, respectively. In addition, unnotched specimens and samples with a 30° notch angle were also tested previously, and the data were published [20]. Here, the data (stress data of unnotched specimens and samples with a 30° notch angle) were directly used and compared with the 60° and 120° notch samples. The tensile results showed that a larger opening angle resulted in a higher tensile strength. This may be due to stress concentration at the notch; a smaller notch angle denoted it being closer to the crack, accelerating the crack growth under load. In Table 1, the stress values of unnotched samples are compared with notched (30° data from [20], 60°, and 120°) samples.

### 3.2. Effect of CB Content on Mechanical Properties of CB/PVDF

#### 3.2.1. Tensile Properties of CB/PVDF Composites

In the beginning of this paper, we reviewed the vital importance of fillers in terms of the mechanical and electrical properties of PVDF matrix composites. The study of Ke et al. [21] showed that, when the filler content of CB in the PVDF/CB system was 1.81 wt.%, the volume resistivity approached 10^13^, and, when the filler was increased to 4 wt.%, the volume resistivity decreased to about 10^3^. Huang et al. [22] found that, with the increase in CB load, the electrical conductivity of composites increased obviously. The fastest growth occurred when the content was 3 wt.%. According to the above literature, the PVDF matrix can exhibit better electrical properties after modification of the CB (<8 wt.%) filler. Based on this, we focused on the mechanical properties of CB/PVDF composites. A series of representative samples were prepared, and the proportion of CB modified filler was 0.5 wt.%, 1 wt.%, 2 wt.%, 4 wt.%, or 6 wt.%. Referring to ASTM D 882-2012, the sample size of the CB/PVDF composite film was 10 mm × 150 mm. For each film, the results were the averaged values of five samples.

The mechanical effect of CB on the PVDF matrix is shown in Figure 4 and Table 2. We found that the tensile strength of the composites increased with the increase in CB content, but not uniformly. When the content of CB was 2 wt.%, the tensile strength of the composite was 39.960 MPa. The strain of the composite material reached the maximum value of 0.4 set by the tensile test. As the weight fraction of CB was increased to 6 wt.%, the tensile strength of the composite decreased to 34.845 MPa, and its strain decreased to 0.071. At the end of the experiment, there was no obvious plastic deformation in the fracture samples, and the fracture was flat and bright. This may have been due to the addition of CB hindering the movement of the PVDF molecular chain, which reduced the elasticity and the elongation of the composite.

The results showed that CB can enhance the mechanical properties of PVDF materials. When CB (2 wt.%) was added into the PVDF polymer, the tensile strength was 139.76% higher than that of the pure PVDF material. Electrical conductivity increased 10-fold and physical resistivity decreased 10-fold [21,22]. However, the addition of CB content could improve the electrical properties, but it reduced the tensile strength of the composite.

#### 3.2.2. SEM and EDS Analysis of CB/PVDF Composites

The dispersion of the filler in a matrix affects the interaction between the polymer and filler, which is very important to the mechanical properties of polymer composites [18]. In order to confirm the composition of the composite material, and to observe the dispersion and mixing of CB particles in the PVDF matrix, SEM observations were carried out on the samples, as shown in Figure 5. The SEM surface micrographs of the samples showed that the surface of the composite film was smooth, and the distribution of particles in the composite film was relatively uniform. In the fracture surface micrographs of Figure 5b, at 20,000× magnification, the section was regular, and some particles were clustered together, but this was not obvious.

Figure 6 shows the EDS diagram of CB/PVDF. Figure 6a–c show the distribution diagram of C, F, and O elements on the surface of the composite film. There was a high proportion of C on the surface of the modified film, in addition to O, F, and other elements. A stable presence of C also confirmed the uniform distribution of CB on the PVDF composite films. The EDS spectrum in the range of 0–10 keV is shown in Figure 6d, because no peaks were detected at energies higher than 10 keV. Samples filled with the nanoparticles showed the characteristic peaks of F, C, and O elements at 0.277 keV, 0.525 keV, and 0.677 keV, respectively. Moreover, the atomic ratio of the CB/PVDF composite in this work detected by EDS is shown in Figure 6d. Because CB conductive tape was used as a carrier in the EDS test, the measured C content was relatively high.

### 3.3. Effect of CB/SiO_2_ on Mechanical Properties of Composites

#### 3.3.1. Effect of CB and SiO_2_ Contents on Tensile Properties of Composites

In this section, all films were fabricated with the same preparation process, and the proportions and fillers were different; CB (0.5 wt.%, 1 wt.%, 2 wt.%, 4 wt.%, 6 wt.%) and SiO_2_ (2 wt.%, 4 wt.%, 6 wt.%) materials with different contents in the PVDF matrix were selected, as shown in Table 3. The test results showed that, when the content of SiO_2_ and CB was 2 wt.%, the tensile strength of CB was the highest, reaching 37.89 MPa. When the SiO_2_ content was 4 wt.% and the CB content was 1 wt.%, the tensile strength of the composite reached 44.76 MPa. When the content of SiO_2_ was 6 wt.% and the CB content was 1 wt.%, the tensile strength was 32.13MPa. It was found that 1 wt.% CB/4 wt.% SiO_2_/PVDF composite had the largest tensile strength and the greatest strain after a series of tests, as shown in Figure 7.

#### 3.3.2. SEM and EDS Analysis of CB/SiO_2_/PVDF Composites

The morphological structure of the composite films was observed by SEM. The surface and cross-section micrographs of the near-surface skin layer are shown in Figure 8 and Figure 9. The SEM surface images (Figure 8) clearly proved the existence of SiO_2_ particles on the surface of SiO_2_/PVDF hybrid films, and the dispersion of SiO_2_ particles was good. With the increase in SiO_2_ content, the distribution of particles on the sample surface obviously increased. A small number of particles were distributed unevenly, but this was not overly apparent. The film showed an asymmetric structure in the cross-section SEM micrographs. In Figure 9b,d, filamentary sections could be seen under 5000× and 50,000× magnification, respectively. The ductility changed well during the tensile test. The nano-SiO_2_ particles on the surface were clear and evenly distributed.

The EDS spectra of the surface are provided in Figure 10. The main peak of F was at 0.677 keV and the strong peak of C was at 0.227 keV. These two elements were abundant because they are the backbone of PVDF. The peak value of Si at 1.740 keV and the peak value of O at 0.525 keV were also detected, indicating the presence of SiO_2_ in the composite. The content of SiO_2_ in Figure 10a,c was 1 wt.% and 6 wt.%, respectively. It can be seen that the distribution of Si and O elements in Figure 10c increased significantly, which was consistent with the results of the energy spectrometry. A steady presence of Si throughout the sample also confirmed the uniform distribution of C and Si across the CB/SiO_2_/PVDF composite film. The EDS results confirmed the presence of SiO_2_ in the composites and met the expected amount.

#### 3.3.3. XRD Analysis of CB/SiO_2_/PVDF Composites

Sardarabadi et al. [16] found that the XRD patterns of pure PVDF films show two obvious peaks at 2*θ* ≈ 18.2548° and 19.9406°, which reflect the characteristics of an *α* crystal. With the addition of the CB filler, the XRD patterns of the PVDF composite film and pure PVDF film are significantly different.

For the PVDF film filled with CB, only one peak appeared at 20.6580°, which was mainly the feature of *a β* crystal. This section mainly studied the polycrystalline state of the PVDF composite film by XRD, as shown in Figure 11. The CB/SiO_2_/PVDF composite films exhibited similar diffraction peaks at 20.687°, corresponding to (2 0 0) planes of PVDF crystals in the *β* phase. There was a weak X-ray diffraction peak at 2*θ* ≈ 38.609° in the PVDF film, which was the characteristic diffraction peak on the crystal faces of (1 1 1) and (2 0 1) of the PVDF β phase. The results showed that PVDF in the prepared CB/SiO_2_/PVDF film was mainly in the polar phase. The addition of CB and SiO_2_ nanoparticles did not lead to the formation of the non-polar *α* crystal phase in the PVDF composite. The main diffraction peaks of the samples were at 20.687°, indicating the existence of the polar *β* phase.

#### 3.3.4. TG Analysis of CB/SiO_2_/PVDF Composites

The thermal properties of the polymer were studied by DTG and TG. The temperature change curves of DTG and TG of the composite materials are shown in Figure 12. The test sample was 9.0682 mg in mass, and the heating rate was 10 °C/min. The results showed that the thermal weight loss temperature of PVDF composites was more than 5% above 430 °C, and the residual mass was 31.81% at 795.83 °C in a nitrogen atmosphere. The data of the TG curve and DTG curve almost corresponded with each other. The thermal degradation temperature of the composite material was high, which indicated that the material had good thermal stability.

## 4. Conclusions

In the process of practical application, cracks and other damage problems are inevitable. The analysis of the fracture damage mechanism of composite materials will help to improve the bearing capacity of materials. In this paper, the fracture damage of composite materials was studied. The relationship between the different notch forms and the mechanical properties of the composite was obtained.

The analysis of the effect of CB on the mechanical properties of the PVDF matrix indicated that the optimal performance was observed when CB content was 2 wt.%. The elongation was more than 40%, and the tensile stress was 11.374 MPa, which was 39.788% higher than that of pure PVDF.

When there were two kinds of modified materials (CB and SiO_2_) in PVDF, an interaction between different fillers was formed. The optimum ratio of the filler content in the tensile mechanical properties of the CB/SiO_2_/PVDF composite was CB:SiO_2_:PVDF = 1:4:95. The related tensile strength was 23.933% higher than 1 wt.% CB/PVDF and 56.580% higher than pure PVDF.

Microscopic characterization, i.e., SEM and EDS, showed that CB and SiO_2_ nanoparticles were well combined within the PVDF matrix. After magnetic stirring and ultrasonic dispersion, there was no obvious agglomeration, which denoted a good strengthening effect on the fracture performance of the PVDF matrix.

Future research should focus on the piezoelectric properties, as well as the mechanical and thermodynamic properties, of polymer (CB/PVDF, SiO_2_/PVDF) composites. Authors should discuss the results and how they can be interpreted relative to previous studies and working hypotheses. The findings and their implications should be discussed in the broadest context possible. Future research directions may also be highlighted.

## Figures and Tables

**Figure 1 polymers-12-00146-f001:**
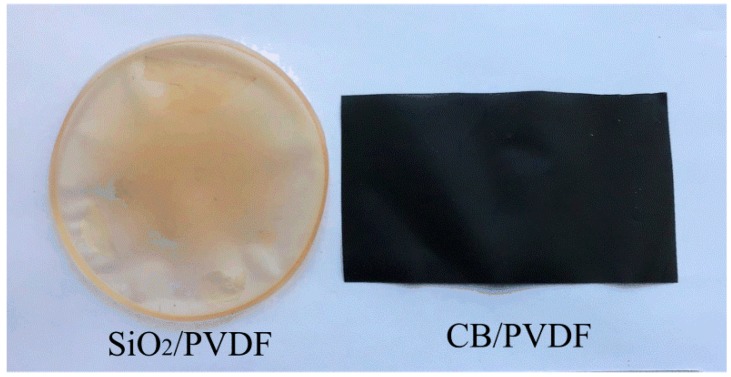
Composite samples.

**Figure 2 polymers-12-00146-f002:**
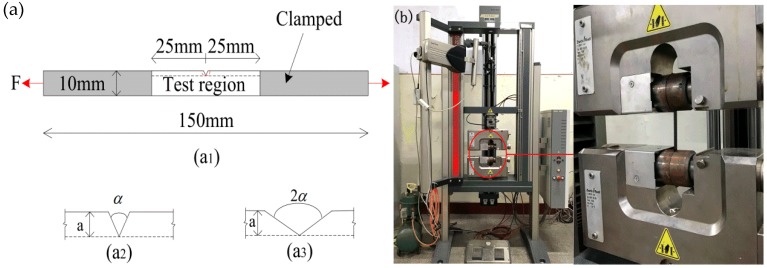
Uniaxial tensile test of composites with different angle notches: (**a**) sample size; (**b**) tensile test.

**Figure 3 polymers-12-00146-f003:**
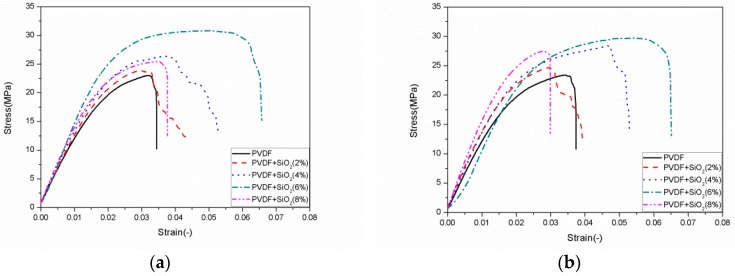
Stress–strain curves of notched samples at different angles: (**a**) 60°; (**b**) 120°.

**Figure 4 polymers-12-00146-f004:**
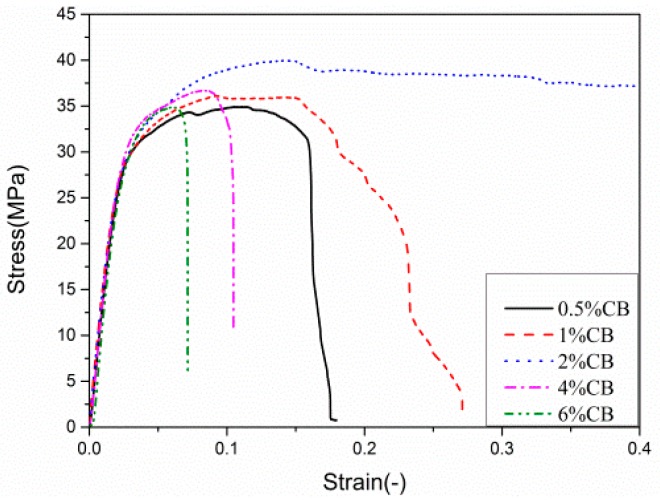
Stress–strain curves of carbon black (CB)/polyvinylidene fluoride (PVDF) composites.

**Figure 5 polymers-12-00146-f005:**
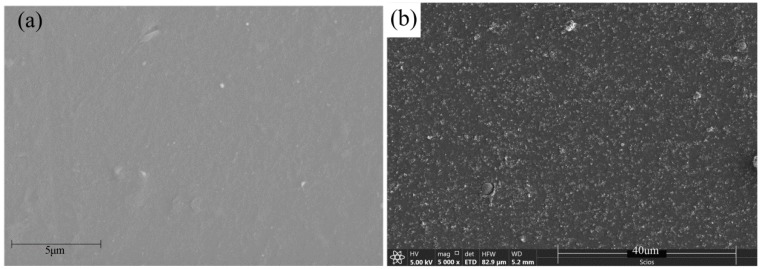
SEM micrographs of CB/PVDF composites (CB content 2.0 wt.%): (**a**) SEM surface micrographs; (**b**) fracture surface micrographs.

**Figure 6 polymers-12-00146-f006:**
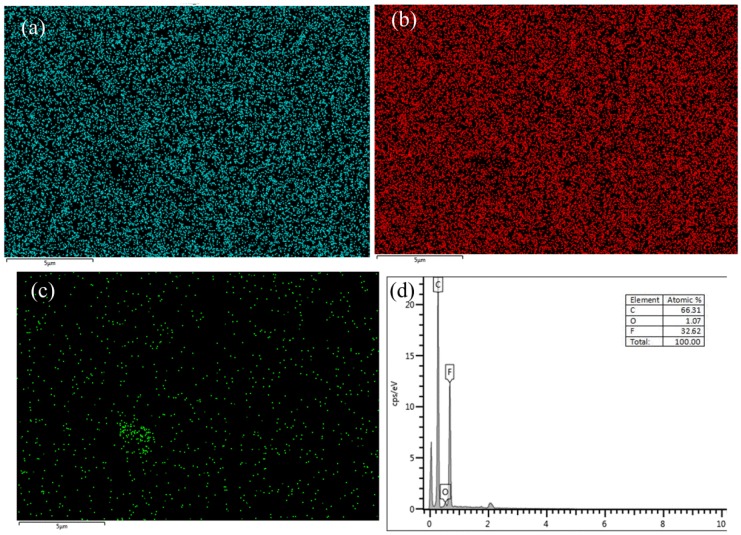
Energy-dispersive spectrometry (EDS) of CB/PVDF composite film (CB content 6.0 wt.%): (**a**) elemental mapping for F; (**b**) elemental mapping for C; (**c**) elemental mapping for O; (**d**) EDS spectrum of CB/PVDF.

**Figure 7 polymers-12-00146-f007:**
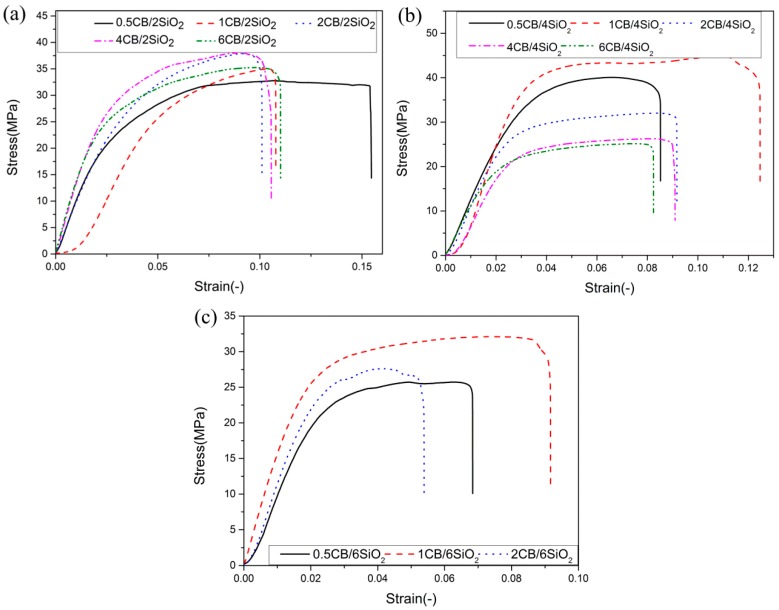
Stress–strain curves of CB/SiO_2_/PVDF composites: (**a**) 2 wt.% SiO_2_; (**b**) 4 wt.% SiO_2_; (**c**) 6 wt.% SiO_2_.

**Figure 8 polymers-12-00146-f008:**
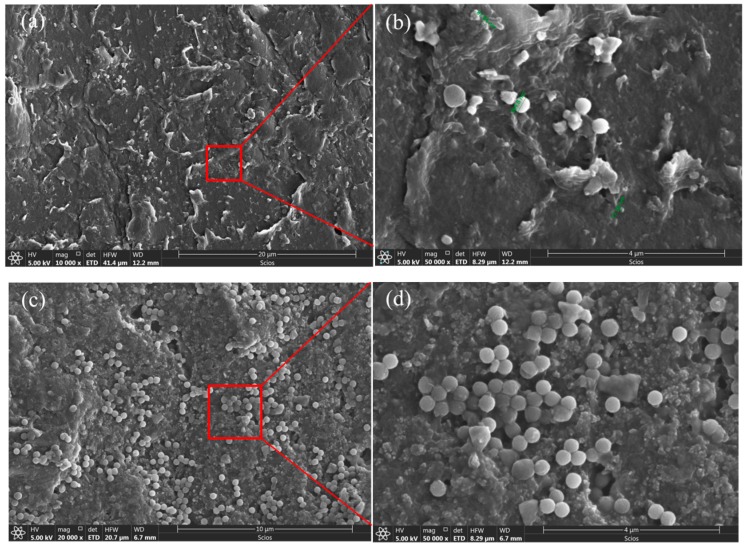
SEM surface micrographs: (**a**) 2 wt.% CB/2 wt.% SiO_2_/PVDF film 10,000×; (**b**) 2 wt.% CB/2 wt.% SiO_2_/PVDF film 50,000×; (**c**) 2 wt.% CB/4 wt.% SiO_2_/PVDF film 20,000×; (**d**) 2 wt.% CB/4 wt.% SiO_2_/PVDF film 50,000×.

**Figure 9 polymers-12-00146-f009:**
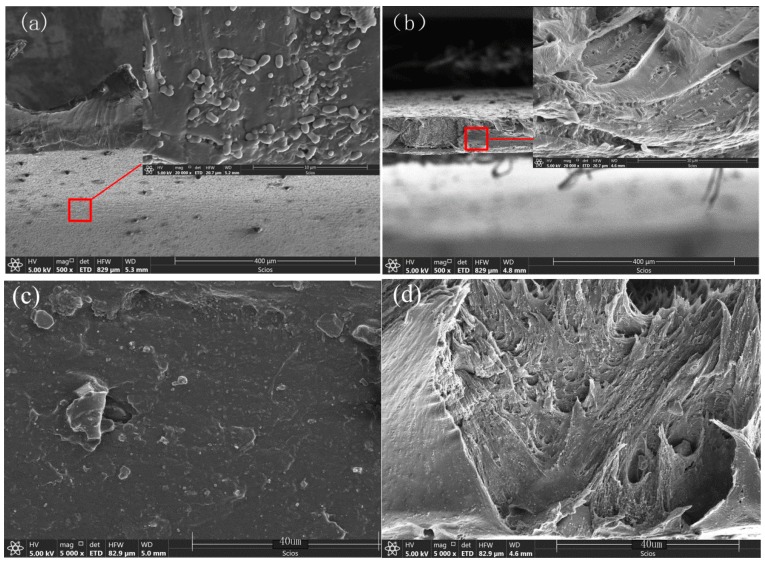
SEM cross-section micrographs of CB/SiO_2_/PVDF composites: (**a**) 2 wt.% CB/6 wt.% SiO_2_/PVDF film; (**b**) 2 wt.% CB/2 wt.% SiO_2_/PVDF film; (**c**) 6 wt.% CB/4 wt.% SiO_2_/PVDF film; (**d**) 0.5 wt.% CB/4 wt.% SiO_2_/PVDF film.

**Figure 10 polymers-12-00146-f010:**
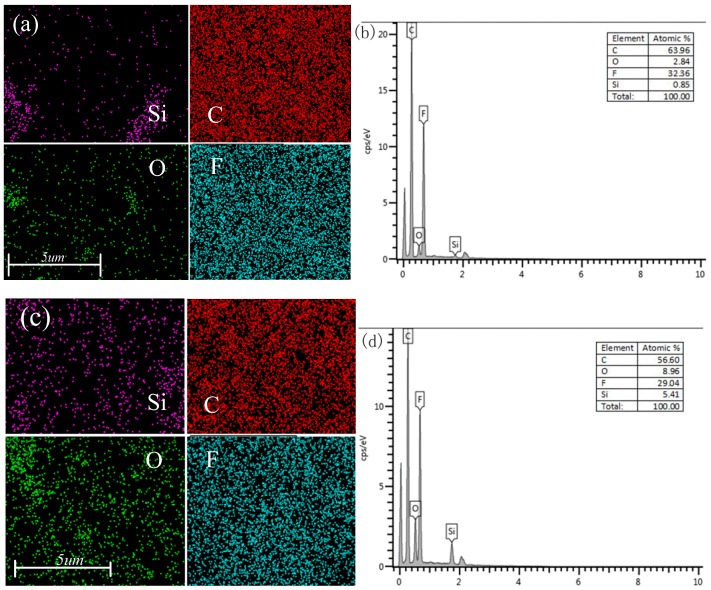
EDS of CB/SiO_2_/PVDF composite film (CB content 2 wt.%): (**a**) elemental mapping for 2 wt.% CB/1 wt.% SiO_2_/PVDF; (**b**) EDS spectrum of 2 wt.% CB/1 wt.% SiO_2_/PVDF; (**c**) elemental mapping for 2 wt.% CB/6 wt.% SiO_2_/PVDF; (**d**) EDS spectrum of 2 wt.% CB/6 wt.% SiO_2_/PVDF.

**Figure 11 polymers-12-00146-f011:**
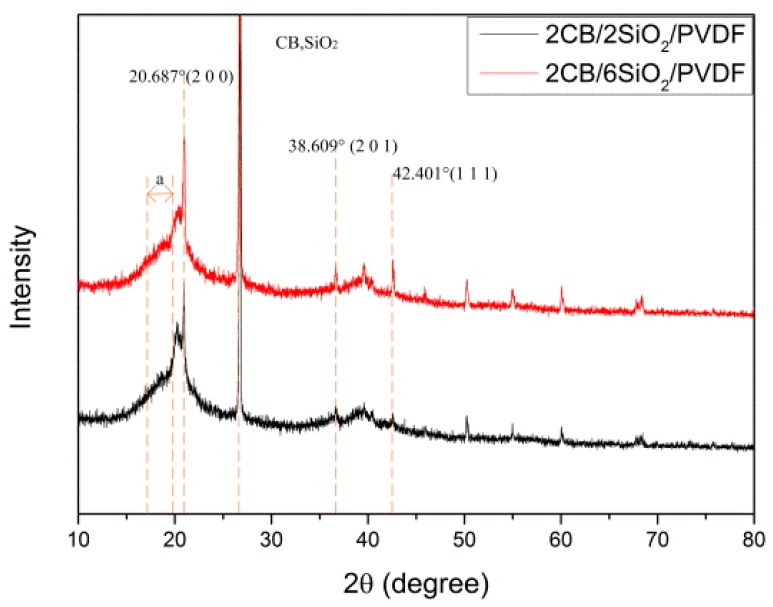
X-ray diffraction (XRD) curve of CB/SiO_2_/PVDF composite film.

**Figure 12 polymers-12-00146-f012:**
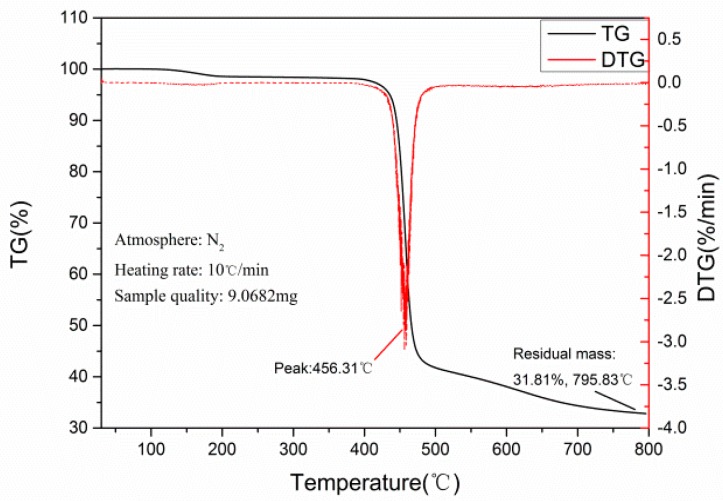
Thermogravimetry (TG) and differential TG (DTG) analysis curves of 1 wt.% CB/4 wt.% SiO_2_/PVDF composite.

**Table 1 polymers-12-00146-t001:** The tensile strength of different notched forms.

SiO_2_ Content (wt.%)	Unnotched Stress (MPa) [20]	30° Stress (MPa) [20]	60° Stress (MPa)	120° Stress (MPa)
0 wt.%	28.594 ± 1.805	21.352 ± 1.782	23.009 ± 2.148	23.406 ± 1.602
2 wt.%	29.598 ± 1.796	22.324 ± 1.224	23.881 ± 1.574	24.671 ± 1.439
4 wt.%	30.147 ± 1.897	22.648 ± 1.935	26.382 ± 2.666	28.385 ± 1.529
6 wt.%	35.872 ± 2.038	25.248 ± 1.731	30.831 ± 1.401	29.718 ± 1.814
8 wt.%	27.828 ± 2.259	21.056 ± 1.679	25.404 ± 2.273	27.451 ± 1.671

**Table 2 polymers-12-00146-t002:** Tensile strength and strain of carbon black (CB)-modified polyvinylidene fluoride (PVDF) composites.

CB Content (wt.%)	Tensile Strength (MPa)	*σ-*Comparison (%)	Strain (-)	*ε*-Comparison (%)
0 wt.%	28.594 ± 1.805	100	0.073 ± 0.007	100
0.5 wt.%	34.923 ± 2.319	122.15	0.178 ± 0.009	243.8356
1 wt.%	36.116 ± 2.843	126.32	0.269 ± 0.014	368.4932
2 wt.%	39.960 ± 1.662	139.76	0.400 ± 0.004	547.9452
4 wt.%	36.703 ± 1.494	128.37	0.104 ± 0.010	142.4658
6 wt.%	34.845 ± 1.812	121.87	0.071 ± 0.003	97.26027

**Table 3 polymers-12-00146-t003:** Information of the samples.

Sample	PVDF (wt.%)	SiO_2_ (wt.%)	CB (wt.%)	Tensile Strength (MPa)
0.5CB/2SiO_2_/PVDF	97.5	2	0.5	32.81
1CB/2SiO_2_/PVDF	97	2	1	34.99
2CB/2SiO_2_/PVDF	96	2	2	37.89
4CB/2SiO_2_/PVDF	94	2	4	37.77
6CB/2SiO_2_/PVDF	92	2	6	35.27
0.5CB/4SiO_2_/PVDF	95.5	4	0.5	40.08
1CB/4SiO_2_/PVDF	95	4	1	44.76
2CB/4SiO_2_/PVDF	94	4	2	32.04
4CB/4SiO_2_/PVDF	92	4	4	25.30
6CB/4SiO_2_/PVDF	90	4	6	25.15
0.5CB/6SiO_2_/PVDF	93.5	6	0.5	25.73
1CB/6SiO_2_/PVDF	93	6	1	32.13
2CB/6SiO_2_/PVDF	92	6	2	27.63

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
