# Peer review of "Comprehensive Analysis of Mechanical Properties of CB/SiO2/PVDF Composites"

_polymers, 2020, doi:10.3390/polym12010146_

Round 1

Reviewer 1 Report

1.       The manuscript has many English language problems, at least on the lines: 43, 47, 50, 51, 54, 58, 60, 88, 92, 128, 146, 170, 174, 219, 229, 230, 281, 317, 343, and 346

2.       Line 55 “A certain amount of CB lubricant can increase or decrease scratch resistance.”??increase or decrease?

3.       Line 60 “strain rate of the brittle toughness ratio is large near the glass transition temperature.”??what is the meaning of this statement?

4.       Line 215 “For each film, the results were the averaged values of five samples.” The results in Table 1 and 2 and Fig.4 should present statistical data, instead of only average, for example: average ± dev.

5.       Fig.4 could be deleted because it is redundant to Table 1 data, Fig. 3 has also shown some representative curves. Labels in Fig.4 are too small.

6.       Line 259 “(d) EDS spectra of SiO2/PVDF”?? Not CB/PVDF?

7.       Notations of some SEM images are too small to be seen, such as Fig.6b, Fig. 10c and d.

8.       Line 288 “(a)” is missing.

9.       Line 295 “The main peak of F is 0.677 keV, and the strong peak of C and O is 0.227 and 0.525 keV. These three elements are abundant because they are the backbone of PVDF.” The three elements of PVDF should be C, H, and F, not O. Please clarify the source of O.

10.    Fig. 11, the scale of EDS images are too small.

11.    Line 323 “The thermal properties of the polymer were studied by DSC and TG.” However, 3.3.4. part only discussed TG results, no DSC work.

12.    Line 349 “The future research should be focused on the mechanical, thermal and electrical properties of piezoelectric polymer (CB/PVDF).” Most of this manuscript is discussing the properties of CB/SiO2/PVDF, and the results are pretty good, why the conclusion about future work is CB/PVDF, instead of CB/SiO2/PVDF?

I look forward to the studies of electrical properties of the composite in future manuscript.

Author Response

The manuscript has many English language problems, at least on the lines: 43, 47, 50, 51, 54, 58, 60, 88, 92, 128, 146, 170, 174, 219, 229, 230, 281, 317, 343, and 346

Re:

Line 43: The composite material is mainly used in building walls, roads and airports. ï¼ˆline 38 in the revision)

Line 47: PVDF has five different crystal types, among which the β crystal type of PVDF has excellent dielectric, piezoelectric and ferroelectric properties.(line 39)

Line 50, 51: Sentences with language problems have been deleted.

Line 54: Brostow W.[13] have optimized the friction and wear properties of PVDF by adding CB nanoparticles. ï¼ˆline 65)

Line 58: Laiarinandrasana L. et al.[17] have analyzed the mechanical behavior of PVDF. Smooth notched specimens were prepared and tested at -100℃-20℃ under different strain rates. ï¼ˆline 79)

Line 60: A larger brittle-ductile area ratio is found in case of higher strain rate and lower temperature. ï¼ˆline 80)

Line 88: Sentences with language problems have been deleted.

Line 92: Sentences with language problems have been deleted.

Line 128: Sentences with language problems have been deleted.

Line 146: Secondly, the mixed solvent was dispersed for 10-15 minutes by ultrasonic vibration machine. ï¼ˆline 108)

Line 170: The thermal properties of the composites were studied by thermogravimetric analysis (TG-DTG) according to ASTM D7426-171 08(2013) standard. The instrument used in the experiment is STA6000 (PerkinElmer, USA). (line 132)

Line 174: In this part, the effect of notch angle on the mechanical properties of the composite is studied. (line 137)

Line 219: When the content of CB is 2 wt%, the tensile strength of the composite is 39.960 MPa. (line 182)

Line 229,330: However, the addition of CB content can improve the electrical properties, but reduce the tensile strength of the composite. (line 194)

Line 281:  The film shows an asymmetric structure in the cross-section micro graphs of SEM. (line 244)

Line 317: The results show that PVDF in the prepared CB/SiO2/PVDF film is mainly polar phase. The addition of CB and SiO2 nanoparticles did not lead to the formation of non-polar α crystal phase in PVDF composite. (line 284)

Line 343: The optimum ratio of the filler content in the tensile mechanical properties of CB/SiO2/PVDF composite is CB: SiO2: PVDF = 1:4:95. The related tensile strength is 23.933% higher than 1CB/PVDF and 56.580% higher than pure PVDF. (line 308, 309)

Line 346: CB and SiO2 nanoparticles are well combined with PVDF matrix. After magnetic stirring and ultrasonic dispersion, there is no obvious agglomeration, which means a good strengthening effect on the fracture performance of PVDF matrix. (line 312)

Line 55 “A certain amount of CB lubricant can increase or decrease scratch resistance.”??increase or decrease?

Re:

The description of “ A certain amount of CB lubricant can increase or decrease scratch resistance. ” in this paper is not accurate enough. Therefore, change the sentence to “Adding a proper amount of CB lubricants (such as 3%) to PVDF composite can reduce the friction of the material. However, less (1% or 1.5%) or more particles will increase the friction of the composite. ”. (Line 66)

We refer to the paper “ Effects of carbon black on tribology of blends of poly(vinylidene fluoride) with irradiated and non-irradiated ultrahigh molecular weight polyethylene[13] ” published by Witold Brostow. Figure 1 in the response letter shows the relationship between the friction results of 70%PVDF+30%UHMWPE and CB concentration. The author analyzes the curve, and the original is on page 5060: “The rule‘put in more additive to get better results’ does not work. The addition of up to 1 phr CB (for dynamic) and up to 1.5 phr (for static) only increases both kinds of friction. However, around 3 phr minima are seen. It is at this concentration that both static and dynamic friction in Fig. 1 begin to fall; carbon black at the surface apparently begins to perform its job of lubrication. However, above 3 phr the friction increase again. ” 

Fig. 1. Friction results for 70% PVDF+30% UHMWPE as a function of

CB concentration.[13]

[13] Brostow, W.; Keselman, M.; Mironi-Harpaz I. Effects of carbon black on tribology of blends of poly (vinylidene fluoride) with irradiated and non-irradiated ultrahigh molecular weight polyethylene [J]. Polymer, 2005, 46(14): 5058-5064.

Line 60 “strain rate of the brittle toughness ratio is large near the glass transition temperature.”??what is the meaning of this statement?

Re:

The description of “ strain rate of the brittle toughness ratio is large near the glass transition temperature. ” in this paper is not comprehensive. Therefore, change the sentence to “A larger brittle-ductile area ratio is found in case of higher strain rate and lower temperature. ”.  (Line 80 and 81)

This sentence mainly refers to the paper “Temperature dependent mechanical behaviour of PVDF: Experiments and numerical modelling” L. Laiarinandrasana. Figure 14 (on page 1315) shows the ratio of brittle-ductile area evolution under different temperatures and (below Tg) and strain rates. The results indicates: “For a given strain rate (vertical line in Fig. 15a), the higher the temperature, the lower the brittle-ductile area ratio. This effect is clearly shown in Fig. 15b where the brittle-ductile area ratio is plotted against the test temperature (parameterised by the strain rate). The abovementioned effects of the temperature and the strain rate are obviously reproduced. This shows that the brittle-ductile area ratio can be considered as a good indicator of the damage and/or fracture mechanisms. Namely, there is an evidence that strain rate has a larger effect around the glass transition temperature.”

Fig. 15. Ratio of brittle-ductile area evolution on smooth bars at temperatures below Tg, with respect to (a) the strain rate (from 1.5×10-5 to 1.5×10-1 /s ), (b) the temperature (from -100 to -50°C).[17]

[17] Laiarinandrasana, L.; Besson, J.; Lafarge, M. Temperature dependent mechanical behaviour of PVDF: experiments and numerical modelling[J].INT. J. PLASTICITY. 2009, 25(7): 1301-1324.

Line 215 “For each film, the results were the averaged values of five samples.” The results in Table 1 and 2 and Fig.4 should present statistical data, instead of only average, for example: average ± dev.

Re:

Data in Tables 1 and 2 is not detailed enough. We have supplemented and amended that in the revised article. The specific data are as follows:  (Line 166 and 196)

Table 1. The tensile strength of different notched forms.

SiO2 content (wt %)

Unnotched

stress(MPa)[20]

30° stress(MPa)[20]

60° stress(MPa)

120° stress(MPa)

0 wt%

28.594±1.805

21.352±1.782

23.009±2.148

23.406±1.602

2 wt%

29.598±1.796

22.324±1.224

23.881±1.574

24.671±1.439

4 wt%

30.147±1.897

22.648±1.935

26.382±2.666

28.385±1.529

6 wt%

35.872±2.038

25.248±1.731

30.831±1.401

29.718±1.814

8 wt%

27.828±2.259

21.056±1.679

25.404±2.273

27.451±1.671

Table 2. Tensile strength and strain of CB modified PVDF composites.

CB content (wt %)

Tensile strength (MPa)

σ-comparison (%)

Strain

(-)

ε-comparison (%)

0 wt%

28.594±1.805

100

0.073±0.007

100

0.5 wt%

34.923±2.319

122.15

0.178±0.009

243.8356

1 wt%

36.116±2.843

126.32

0.269±0.014

368.4932

2 wt%

39.960±1.662

139.76

0.400±0.004

547.9452

4 wt%

36.703±1.494

128.37

0.104±0.010

142.4658

6 wt%

34.845±1.812

121.87

0.071±0.003

97.26027

Fig.4 could be deleted because it is redundant to Table 1 data, Fig. 3 has also shown some representative curves. Labels in Fig.4 are too small.

Re:

Most of the data in Figure 4 and Table 1 are duplicate, so Figure 4 have been deleted in the revised version.

Line 259 “(d) EDS spectra of SiO2/PVDF”?? Not CB/PVDF?

Re:

The name in Figure 5(d) should be “(d) EDS spectra of CB/PVDF”. Errors have been corrected in the revision on line 221.

Notations of some SEM images are too small to be seen, such as Fig.6b, Fig. 10c and d.

Re:

We use the drawing software (Microsoft Visio) to redraw and mark the relevant dimensions of Figure 6b and figure 10c and d. The modified picture is as follows:

Figure 5. SEM micrographs of CB/PVDF composites (CB content 2.0 wt %): (a) SEM surface micrographs; (b) fracture surfaces micrographs.

Figure 6(original article) ➡ Figure 5(Revised draft)

Figure 9. SEM cross-section micrographs of CB/SiO2/PVDF composites: (a) 2CB/6SiO2/PVDF film; (b) 2CB/2SiO2/PVDF film; (c) 6CB/4SiO2/PVDF film; (d) 0.5CB/4SiO2/PVDF film.

Figure 10(original article) ➡ Figure 9(Revised draft)

Line 288 “(a)” is missing.

Re:

The missing (a) has been added to the revised manuscript, on line 250.

Figure 8. SEM surface micrographs: (a)2CB/2SiO2/PVDF film 10000x; (b) 2CB/2SiO2/PVDF film 50000x; (c) 2CB/4SiO2/PVDF film 20000x; (d) 2CB/4SiO2/PVDF film 50000x.

Line 295 “The main peak of F is 0.677 keV, and the strong peak of C and O is 0.227 and 0.525 keV. These three elements are abundant because they are the backbone of PVDF.” The three elements of PVDF should be C, H, and F, not O. Please clarify the source of O.

Re:

PVDF materials contain three elements: C, H, F, and do not contain O. SiO2 particles were added to the composite during the preparation. The main components of SiO2 are Si and O. Therefore, the presence of O element is observed in EDS test results.            

There is an error in this article. Now, modify it as follows: “The EDS spectra of the surface are provided in Figure 10. The main peak of F is 0.677keV and the strong peak of C is 0.227. These two elements are abundant because they are the backbone of PVDF. The peak value of Si (1.740 keV) and the peak value of O (0.525 keV) are also detected, indicating the presence of SiO2 in the composite.”

Fig. 11, the scale of EDS images are too small.

Re:

We have enlarged the images in Fig. 11 (a) and (c). The distribution of elements can be observed more clearly than the original article.

Figure 10. EDS of CB/SiO2/PVDF composite film (CB content 2wt%): (a) elemental mapping for 2CB/1SiO2/PVDF; (b) EDS spectra of 2CB/1SiO2/PVDF; (c) elemental mapping for 2CB/6SiO2/PVDF; (d) EDS spectra of 2CB/6SiO2/PVDF.

Figure 11(original article) ➡ Figure 10(Revised draft)

Line 323 “The thermal properties of the polymer were studied by DSC and TG.” However, 3.3.4. part only discussed TG results, no DSC work.

Re:

In section 3.4.4, we give the TG-DTG curve of the composite (Figure 12) and analyze it, not DSC test. Therefore, correct the error as follows: “The thermal properties of the polymer are studied by DTG and TG. The temperature change curves of DTG and TG of composite materials are shown in Figure 12. ”

Figure 12. TG and DTG analysis curves of 1CB/4SiO2/PVDF composite.

Line 349 “The future research should be focused on the mechanical, thermal and electrical properties of piezoelectric polymer (CB/PVDF).” Most of this manuscript is discussing the properties of CB/SiO2/PVDF, and the results are pretty good, why the conclusion about future work is CB/PVDF, instead of CB/SiO2/PVDF?

Re:

The mechanical properties of CB/PVDF composite are greatly improved compared with pure PVDF, and its piezoelectric effect is also a property that has attracted much attentions [4-12]. The addition of SiO2 nanoparticles is mainly to enhance the mechanical properties of PVDF composite[3,19]. Of course, the influence of CB and SiO2 nanoparticles on the properties of composite and its application in engineering should be considered and studied.

  In the revised article, we revised this issue as follows: “The future research should focus on the piezoelectric properties, mechanical and thermodynamic properties of polymer (CB/PVDF, SiO2/PVDF) composites.”

Reviewer 2 Report

According to the Abstract the paper presents the mechanical properties of CB/SiO2/PVDF Composites based on different procedures.

Please, specify in the abstract what is the aim and the novelty of this work.

After carefully reviewing this paper, I recommend that it:

On page 2 line 67 you introduce the “ASI value”, please tell us what this ASI value represents.

You spoke in the introduction of several terms through abbreviations, please explain what it represents each term introduced.

In my opinion the Introduction is too bushy, too many explanations, very difficult to follow and understand this chapter.

From my point of view the points a) and b) in figure 1 are not scientifically relevant, except point c).

On page 4 line 176 you talked about “The notch angle of the tested material is 60° and 120°” and then on page 5 in table 1 you offered data for four types of samples including unnotched sample and 30o notched.

Then in line 197 pages 5 you talked about “Some experimental data can be found in literature [19].” please specify which experimental data you are talking about?

The content of the head of table 1 is somewhat ambiguous ... = write 30 degrees and in brackets indicate the unit of measure (MPa). If you look directly only at this table his understanding is somewhat ambiguous.

In this conditions figure 4 is redundant, more precisely express graphically what you say in Table 1.

I think it was interesting when studying the mechanical properties to be occupied with the behavior of wear, elasticity, hardness, adhesion, etc. not only tensile strength.

The work is difficult to follow because you lose in these all details.

Please specify more clearly the motivation that led to this work. The numerous measurements are to be appreciated, but the purpose of this work must be better scored.

More elaborate and clear conclusions should be given, including comparisons with the literature.

None of methods can be considered original, nor are the motivations and goals of the experimental efforts very well provided to the reader. Nonetheless, the paper can be of interest for the audience of Polymers, mostly because it may represent an additional (compared to the very many similar papers appeared in the last decades).

But it is absolutely necessary to highlight the purpose of the work and its motivation, as well as to highlight the clear contribution of the authors in comparison with the specialized literature.

Author Response

According to the Abstract the paper presents the mechanical properties of CB/SiO2/PVDF Composites based on different procedures.

Please, specify in the abstract what is the aim and the novelty of this work.

Re:

We revised the abstract of the article as follows:

Abstract: Damage is a key problem that limits the application of polymer membranes. In this paper, conductive carbon black (CB) and silicon dioxide (SiO2) reinforced polyvinylidene fluoride (PVDF) composites are prepared by solution mixing method. Through uniaxial tensile test, the fracture and damage characteristics of the material are analyzed. When the structure has inevitable notch damage, changing the notch angle is very helpful for the material to bear more load. In addition, when there are two kinds of fillers in PVDF matrix at the same time, there will be interaction between particles. The microstructure of the composite was characterized by scanning electron microscopy (SEM), energy dispersive spectrometer (EDS) and thermogravimetric (TG). The experimental results indicate that when the ratio of CB: SiO2: PVDF is 1:4:95, the general mechanical properties of the composite are the best.

After carefully reviewing this paper, I recommend that it:

On page 2 line 67 you introduce the “ASI value”, please tell us what this ASI value represents.

Re:

“ASI value” is not explained clearly in the revised version, and it is modified as follows: averaged strain interval (ASI). (line 46)

[4] Xu, H.; Zeng, Z.; Wu, Z. Broadband dynamic responses of flexible carbon black/poly (vinylidene fluoride) nanocomposites: A sensitivity study[J].COMPOS. SCI. TECHNOL. 2017, 149: 246-253.

You spoke in the introduction of several terms through abbreviations, please explain what it represents each term introduced.

Re:

In the revised version, the unexplained part is modified as follows:

Line 51: polyvinylidene fluoride-hexafluoropropylene (PVDF-HFP)

“Liangke Wu’s[5] results showed that when the optimum CB content is 0.5 wt%, the optimum output voltage and power density of the composite film are 204% and 464% of the original PVDF-HFP (hexafluoropropylene) film.”

Line 81: multi-walled carbon nanotubes( MWCNTs)

“ Kai Ke et al. [18] prepared multi-walled carbon nanotubes(MWCNTs)/CB/PVDF composites by melt mixing. ”

Original article: carbon nanotubes(CNT)/CB/PVDF

“ Li Li et al. [16] results showed that the permeability threshold of carbon nanotubes(CNT)/CB/PVDF nano-composites (1.25% CNT) was lower than that of CNT/PVDF (>2.08% CNT). ”

In the revised version, this sentence has been deleted.

Original article: polyvinylidene fluoride/poly(methyl methacrylate) (PVDF/PMMA)  

“ It is found that the addition of CB resulted in the phase fluctuation of PVDF/poly(methyl methacrylate)PMMA blends and the uneven distribution of CB particles in the blends, which greatly reduced the penetration threshold of PVDF/PMMA/CB Composites. ”

In the revised version, this sentence has been deleted.

In my opinion the Introduction is too bushy, too many explanations, very difficult to follow and understand this chapter.

Re:

We have revised the introduction part, as follows:

Introduction

Compared with ordinary materials, composite materials have many characteristics, which can improve or overcome the weakness of a single material, give full play to the advantages of each material. The optimum performance design of materials can be carried out according to the functional requirements. Commonly used composite matrices are metal matrix, inorganic material matrix and polymer matrix. In contrast, polymeric materials own good formability and are light-weight, which is a great promise for applications. Polyvinylidene fluoride (PVDF) is a typical polymer material and widely used in various fields of engineering application with its excellent physical properties. Firstly, it has good thermal and mechanical properties. At a certain pressure and temperature, it can still maintain good mechanical strength. Mainly used for valves, pipes, heat exchangers. Secondly, PVDF has a good chemical stability and can resist the corrosion of most organic solvents, inorganic acids and aromatic hydrocarbons. It can be used for storage and transportation of highly purified chemicals. Thirdly, PVDF has strong weather resistance, high anti-fading and anti-ultraviolet performance in harsh environment. The composite material is mainly used in building walls, roads and airports. Finally, PVDF has five different crystal types, among which the β crystal type of PVDF has excellent dielectric and piezoelectric properties. Mainly used for ultrasonic measurement, pressure sensor and detonation.

In the study of modified composites, carbon black (CB) has become one of the main fillers to improve the piezoelectric and mechanical properties of composites. Batteries, sensors and piezoelectric films all need composite materials with more stable storage, sensing and other functions [1-3]. Hao X. [4] studied the dynamic piezoresistance of the flexible nanocomposite strain sensor made of CB nanoparticles and PVDF composites. In the time domain analysis, the averaged strain interval (ASI) value is less than 0:8 mε, which indicates that CB/PVDF sensor can detect the extremely weak strain related to structural damage, high frequency vibration and ultrasonic, and the excellent sensing accuracy and response frequency of CB/PVDF sensor are related to the microstructure of nanoparticles. Liangke W.’s[5] results showed that when the optimum CB content is 0.5 wt%, the optimum output voltage and power density of the composite film are 204% and 464% of the original PVDF- hexafluoropropylene (HFP) film. The addition of excessive CB will lead to smaller crystal size and lower crystallinity, thus reducing the piezoelectric properties of PVDF-HFP/CB composite films. Lazarraga M. G.[6] have prepared lithium battery anode composite with LiMn2O4 as electrochemical active material, CB as conductor and PVDF as binder. When the CB content is 2wt%, the conductivity increases with the increase of percolation process. The composite exhibits elastic behavior at stress of 6 MPa. Fuertes A.[7] and Ha S. [8] have prepared electrode composites with PVDF, Silicon dioxide(SiO2) as matrix, CB and LiMn2O4 as conductor. The influence of CB content on relative capacitance, energy density and power density is studied. Adding CB nanoparticles to PVDF matrix can improve its dielectric properties[9,10], reduce the penetration threshold of the composite, and improve the conductivity and mechanical properties of the composite [11,12].

The addition of CB nanoparticles can not only improve the piezoelectric properties, but also change the micro properties of the composite. The internal structure of the composite is more compact and the composite has corrosion resistance and wear resistance. Brostow W.[13] have optimized the friction and wear properties of PVDF by adding CB nanoparticles. Adding a proper amount of CB lubricants (such as 3%) to PVDF composite can reduce the friction of the material. However, less (1% or 1.5%) or more particles will increase the friction of the composite. Antunes R. A.[14] have studied the corrosion resistance of CB - synthetic graphite (SG) - PVDF composites. It is found that the corrosion resistance decreases when the CB content increases to 5 wt%. The highest conductivity was achieved for the composition CB = 5 wt%, PVDF = 15 wt%, SG = 80 wt%. Thomas R.[15] studied the effects of CB temperature, soaking time, polymer solution concentration and casting PVDF film thickness on the film properties. A new type of film with uniform surface structure and asymmetric pore structure was obtained. Sardarabadi H.[16] prepared CB/PVDF composites with different mass fractions. It is found that the addition of CB filler made the membrane structure more compact, lower permeation flux and higher contact angle and pervaporation separation index.

In engineering applications, the mechanical properties of composite materials are widely concerned and deeply studied by scholars. Laiarinandrasana L.[17] have analyzed the mechanical behavior of PVDF. Smooth notched specimens were prepared and tested at -100°C-20°C under different strain rates. A larger brittle-ductile area ratio is found in case of higher strain rate and lower temperature. Ke K. [18] prepared multi-walled carbon nanotubes (MWCNTs)/CB/PVDF composites by melt mixing. It is found that the nucleation efficiency of MWCNTs is higher than that of CB for the crystallization of PVDF. Dynamic mechanical analysis shows that MWCNTs have a greater contribution to the storage modulus of PVDF, especially at low temperature. Haddadi S. A.[19] studied the effect of different nano SiO2 content on the mechanical and thermodynamic properties of PVDF. The results show that the tensile strength of the composite is 450% higher than that of PVDF when the fiber diameter is 125-350 nm and the SiO2 content is 2 wt%.

Many researchers have studied the mechanical properties of PVDF and its composites, including modulus of elasticity, viscoelasticity, ultimate stress and strain. However, in the practical application of film materials, fracture damage is inevitable. In this paper, different kinds of  single-edge notched tensile tests are carried out to investigate the damage mechanism of PVDF matrix composites prepared in the laboratory. The fracture damage characteristics of the materials are systematically studied by means of scanning electron microscopy (SEM), X-ray diffraction (XRD) and energy dispersive spectrometer (EDS).

 From my point of view the points a) and b) in figure 1 are not scientifically relevant, except point c).

Re:

Figure 1(a) and (b) are not related to science and do not need to appear in the article. Therefore, the figure has been deleted and modified, as follows:

Figure 1. Composite samples.

 On page 4 line 176 you talked about “The notch angle of the tested material is 60° and 120°” and then on page 5 in table 1 you offered data for four types of samples including unnotched sample and 30onotched.

Then in line 197 pages 5 you talked about “Some experimental data can be found in literature [19].” please specify which experimental data you are talking about?

Re:

From Table 1, the ultimate stress of single notch tensile test with different contents and different angles was listed. The experimental conditions of composite samples are the same, the notch angles are 60° and 120° respectively. In addition, the unnotched specimens and samples with 30° notch angle have also been tested previousl, and the data have been published in my article[20]. Here, the data (stress data of unnotched specimens and samples with 30° notch angle) are directly used and compared with 60° and 120° notch samples. Tensile results show that the larger the opening angle is, the higher the tensile strength is. It may be due to stress concentration at the notch tip; the smaller the notch angle is, the closer it is to the crack shape, and the faster the crack grows under load. In table 1, the stress of a unnotched samples is compared with notched (30°[20], 60°, and 120°) samples.

Reference [19](original article) ➡ Reference [20](Revised draft)

[20] Kong, F. Y.; Chang, M. Z,; Wang, Z. Q. Investigation of the effect of nanosilica on mechanical, structural, and fracture toughness of polyvinylidene fluoride films[J]. Materials Research Express, 2019, 6(10): 105369.

 The content of the head of table 1 is somewhat ambiguous ... = write 30 degrees and in brackets indicate the unit of measure (MPa). If you look directly only at this table his understanding is somewhat ambiguous.

Re:The content of the head of Table 1 is somewhat ambiguous. There are many shortcomings, and the error will be corrected.

Table 1. The tensile strength of different notched forms.

SiO2 content (wt %)

Unnotched

stress(MPa)[20]

30°

stress(MPa)[20]

60° stress(MPa)

120° stress(MPa)

0 wt%

28.594±1.805

21.352±1.782

23.009±2.148

23.406±1.602

2 wt%

29.598±1.796

22.324±1.224

23.881±1.574

24.671±1.439

4 wt%

30.147±1.897

22.648±1.935

26.382±2.666

28.385±1.529

6 wt%

35.872±2.038

25.248±1.731

30.831±1.401

29.718±1.814

8 wt%

27.828±2.259

21.056±1.679

25.404±2.273

27.451±1.671

In this conditions figure 4 is redundant, more precisely express graphically what you say in Table 1.

Re: 

In the revised manuscript, figure 4 is deleted. We have a more detailed description of the data in Table 1.

 I think it was interesting when studying the mechanical properties to be occupied with the behavior of wear, elasticity, hardness, adhesion, etc. not only tensile strength.

The work is difficult to follow because you lose in these all details.

Please specify more clearly the motivation that led to this work. The numerous measurements are to be appreciated, but the purpose of this work must be better scored.

Re:

In this paper, uniaxial tensile test was carried out to study the effect of CB and SiO2 nanoparticles on the tensile strength of PVDF composite. At the same time, the fracture mechanism of the composite was investigated. It is inevitable that cracks and other defects will appear in the actual application. The modification of the form and angle of the cracks will help to change the bearing capacity of the composite. In addition, we found that CB and SiO2 nanoparticles have stress enhancement effect on PVDF composite. However, two kinds of particles exist in the composite at the same time, which will produce the interaction between particles. CB/PVDF is a kind of piezoelectric material that scholars often study. The increase of its mechanical properties is also helpful to its application in strain gauge and other engineering. The tensile strength of CB/SiO2/PVDF is higher than that of CB (2 wt%)/PVDF and SiO2 (6 wt%) / PVDF.

More elaborate and clear conclusions should be given, includingcomparisons with the literature.

Re: 

Conclusions

In the process of practical application, cracks and other damage problems are inevitable. The analysis of fracture damage mechanism of composite materials will help to improve the bearing capacity of materials. In this paper, the fracture damage of composite materials is studied. The relationship between the different notch forms and the mechanical properties of the composite is obtained.

The analysis of the effect of CB on the mechanical properties of PVDF matrix indicatesthat: the optimal performance is observed when CB content is 2wt%. The elongation is more than 40%, and the tensile stress is 11.374 MPa, whichis 39.788% higher than that of pure PVDF. When there are two kinds of modified materials (CB and SiO2) in PVDF, the interaction between different fillers will be formed. The optimum ratio of the filler content in the tensile mechanical properties of CB/SiO2/PVDF composite is CB: SiO2: PVDF = 1:4:95. The related tensile strength is 23.933% higher than 1CB/PVDF and 56.580% higher than pure PVDF. Microscopic characterization, i.e., SEM and EDS show that: CB and SiO2nanoparticles are well combined with PVDF matrix. After magnetic stirring and ultrasonic dispersion, there is no obvious agglomeration, which means a good strengthening effect on the fracture performance of PVDF matrix.

The future research should focus on the piezoelectric properties, mechanical and thermodynamic properties of polymer (CB/PVDF, SiO2/PVDF) composites. Authors should discuss the results and how they can be interpreted in perspective of previous studies and of the working hypotheses. The findings and their implications should be discussed in the broadest context possible. Future research directions may also be highlighted.

None of methods can be considered original, nor are the motivations and goals of the experimental efforts very well provided to the reader. Nonetheless, the paper can be of interest for the audience of Polymers, mostly because it may represent an additional (compared to the very many similar papers appeared in the last decades).But it is absolutely necessary to highlight the purpose of the work and its motivation, as well as to highlight the clear contribution of the authors in comparison with the specialized literature.

Re:

Thank you very much for taking the time for review my article. These valuable opinions are of great help to my article. The quality of the revised paper has been greatly improved. Especially the research purpose of the paper, this is the problem we need to focus on.
